

# Technical note: Studying Li-metaborate fluxes and low-temperature combustion/high-temperature extraction systematics with a new, fully automated *in situ* cosmogenic [14]C processing system at PRIME Lab

Nathaniel Lifton[1,2], Jim Wilson[3], Allie Koester[1]

[1]Department of Earth, Atmospheric, and Planetary Sciences, Purdue University, 550 Stadium Mall Drive, West Lafayette, Indiana, 47907, USA.
[2]Department of Physics and Astronomy and Purdue Rare Isotope Measurement Laboratory (PRIME Lab), Purdue University, 525 Northwestern Avenue, West Lafayette, Indiana, 47907, USA.
[3]Aeon Laboratories, LLC, 5835 North Genematas Drive, Tucson, Arizona, 85704, USA.

*Correspondence to*: Nathaniel Lifton (nlifton@purdue.edu)

**Abstract.** Extraction procedures for *in situ* cosmogenic [14]C (*in situ* [14]C) from quartz require quantitative isotopic yields while maintaining scrupulous isolation from atmospheric/organic [14]C. These time- and labor-intensive procedures are ripe for automation; unfortunately, our original automated *in situ* [14]C extraction and purification systems, reconfigured and retrofitted from our original systems at the University of Arizona, proved less reliable than hoped. We therefore installed a fully automated stainless-steel system (except for specific glass or fused-quartz components) incorporating more reliable valves and improved actuator designs, along with a more robust liquid nitrogen distribution system. As with earlier versions, the new system uses a degassed Li-metaborate ($LiBO_2$) flux to dissolve the quartz sample in an ultra-high-purity oxygen atmosphere, after a lower-temperature combustion step to remove atmospheric/organic [14]C.

We compared single-use high-purity $Al_2O_3$ vs. reusable 90%Pt/10%Rh (Pt/Rh) sample combustion boats. The Pt/Rh boats heat more evenly than the $Al_2O_3$, reducing procedural blank levels and variability for a given $LiBO_2$ flux. This lower blank variability also allowed us to trace progressively increasing blanks to specific batches of fluxes from our original manufacturer. Switching to a new manufacturer returned our blanks to consistently low levels on the order of $(3.4 \pm 0.9)$ x $10^4$ [14]C atoms.

We also analyzed the CRONUS-A intercomparison material to investigate sensitivity of extracted [14]C concentrations to the temperature and duration of the combustion and extraction steps. Results indicate that 1-hr combustion steps at either 500 or 600°C yield results consistent with the consensus value of Jull et al. (2015), while 2 hr at 600°C results in loss of ca. 9% of the high-temperature [14]C inventory. Results for 3 hr extractions at temperatures ranging from 1050°C to 1120°C and 4.5 hr at 1000°C yielded similar results that agreed with the nominal value as well as with published results from most laboratories. On the other hand, an extraction for 3 hr at 1000°C was judged to be incomplete due to a significantly lower measured concentration. Based on these results, our preferred technique is now combustion for 1 hr at 500°C followed by a 3 hr extraction at 1050°C. Initial analyses of the CoQtz-N intercomparison material at our lab yielded concentrations ca. 60% less than those of CRONUS-A, but more analyses of this material from this and other labs are clearly needed to establish a consensus value.



## 1 Introduction

Extracting in situ cosmogenic $^{14}C$ (*in situ* $^{14}C$) from quartz is challenging in that minute quantities of $^{14}C$ must be
extracted and purified from quartz samples while preventing contamination by ubiquitous atmospheric/organic $^{14}C$.
These extraction and purification procedures are time-consuming and labor-intensive when done manually – as such
they are attractive targets for automation. Lifton et al. (2015) presented results from the initial automated *in situ* $^{14}C$
extraction and purification systems at the Purdue Rare Isotope Measurement Laboratory (PRIME Lab), reconfigured
and retrofitted from our original glass systems at the University of Arizona. As hoped, the automation of key
components of our *in situ* $^{14}C$ lab indeed led to increased throughput and reproducibility. While the overall timeline
of the extraction, purification, and graphitization was still ca. 3 days, a single person was able to operate both
automated extraction systems, the automated purification system, and the manual graphitization system
simultaneously, boosting sample throughput significantly over the purely manual systems.

These automated systems comprised two independent extraction systems and a separate $CO_2$ purification system. A
separate system for converting $CO_2$ to graphite was not automated. This required custom design and implementation
of equipment to automate three key aspects of the systems: servo-based valve actuators, temperature control for
cryogenic gas purification, and liquid nitrogen (LN) transfer. While these automated systems improved throughput
over our original purely manual systems, they also required manual transfer of sample gas between separate
extraction, purification, and (manual) graphitization systems.

However, in terms of overall reliability of operation, the limitations of retrofitting our original designs ultimately
became apparent. For example, the glass high-vacuum valves are not precision components – no two are precisely
the same. The valve actuators thus had to adapt to differences in resistance to motion arising from variations in valve
stem and valve bore diameters, as well as to different lengths of travel to adequately seat each valve. As originally
designed, the valve actuators accommodated these variations well, but the mechanical settings at which each
operated properly tended to creep over time, such that sometimes during active processes individual valves might
not indicate that they are closed or open, or might indicate a closed position but not be seated properly and allow
leakage across the valve. Without actively checking on the system status when this happened, the sample gas could
be pumped away by accident, or a process could be interrupted (which could lead to system damage).

Similarly, the LN distribution system in that system was ultimately problematic. LN was transferred from a
pressurized 200 L supply dewar through insulated Teflon tubing to fill dewars on various cold traps. Filling and
emptying of individual dewars was controlled using LN level sensors comprising three resistors in series, positioned
with resistors at empty, nominal, and full levels within each dewar. During the processes, certain cold traps needed
to be alternately filled and emptied. Dewars stationed on those traps were emptied using a small shop vacuum
cleaner via a drain manifold fitted with cryogenic solenoid valves. Particularly at times of high humidity (not as
much of an issue in arid Arizona as in Indiana), ice condensation in those dewars could cause the drain tubing to
clog and interrupt the process sequence. Also, sometimes when a particular dewar was filled and emptied multiple
times in a process sequence, the resistor string would not register the proper voltage during a fill cycle to trigger
shutoff of LN flow, and the dewar would overflow continuously unless an operator was present to close the main





supply dewar valve manually. Thus, although sample throughput and repeatability was considerably improved over manual operation, system reliability was not at the point where one could generally leave a system in unattended operation.

We thus recently purchased and installed a customized Carbon Extraction and Graphitization (CEGS) system from Aeon Laboratories, LLC, similar to that of Goehring et al. (2019) at Tulane University. The largely stainless-steel system (except for specific sections requiring glass or fused-quartz components) incorporates more reliable valves

and improved actuator designs compared to our original system, as well as a robust and efficient liquid nitrogen distribution system. The new system, controlled by a flexible and extensible modular software package written in C#, follows a similar procedure to that of Lifton et al. (2015), using a degassed Li-metaborate (LiBO$_2$) flux to dissolve the quartz sample in a Research Purity (RP) O$_2$ atmosphere. In addition, all sections of the new system are connected, so that one can extract all evolved carbon species as CO$_2$ from a quartz sample, purify and precisely

measure the resulting gas yield, and convert the CO$_2$ to graphite for AMS analysis – all without human intervention. Below we describe key differences relative to the system of Goehring et al. (2019), then present baseline results from the now fully operational system, including procedural blanks and analyses of established intercomparison materials, for both our original single-use high-purity aluminum oxide and new reusable 90%Pt/10%Rh sample boats.

**2 Purdue CEGS design and operation**

The Purdue CEGS (PCEGS) comprises three main modules: two extraction modules and a collection/purification/graphitization module (main CEGS module) (Fig. 1), following the general design of Goehring et al. (2019) but with an additional extraction module. However, the PCEGS differs from the latter system in two key aspects. First, the two PCEGS extraction modules (Tube Furnace 1 [TF1] and Tube Furnace 2 [TF2]) are

connected in parallel, each accommodating a high-temperature resistance furnace with a mullite furnace tube and evacuated by separate vacuum systems distinct from the main CEGS vacuum system (Fig. 1). On the other hand, the Goehring et al. (2019) system comprises one tube furnace in series with the CEGS, evacuated by a single vacuum system. Our design allows each PCEGS extraction module to run processes independently of those controlled by the other modules, enabling increased flexibility in system operations. The other key difference is that condensable

gases evolved on the PCEGS during an extraction procedure in either furnace are trapped in a compact borosilicate glass coil trap held at LN temperature (-196°C) instead of the variable temperature trap (VTT) used for this purpose on the Goehring et al. (2019) system (Figs. 1, 2). The compact coil trap (ca. 3.5 cm diameter x ca. 10 cm tall) derives from our previous larger coil trap designs (e.g., Lifton et al., 2001; Pigati et al., 2010; Lifton et al., 2015), which consistently demonstrated quantitative trapping of minute CO$_2$ quantities from O$_2$ carrier gas. This compact

design ensures similarly reliable CO$_2$ trapping through a 9 mm o.d. x 7 mm i.d. inlet downtube delivering process gases directly to the base of the trap before passing through a constriction connecting the downtube to a 6 mm o.d. x 4 mm i.d. coiled section and outlet tube (Fig. 2). The total length of the trap submerged in LN when operating is ca. 55 cm (ca. 5 cm of the downtube and the ca. 50 cm coil).



Other than incorporating a U-shaped secondary oxidation furnace (9 mm o.d. x 7 mm i.d. filled with 2 mm quartz beads, held at ca. 900°C) from the Lifton et al. (2015) system instead of an inline granular quartz oxidation furnace of the Goehring et al. (2019) design, the rest of the PCEGS utilizes similar hardware to that of the latter. However, our six-reactor graphitization manifold is configured in front of the main purification and measurement process path to achieve a shorter footprint than the linear configuration of the Tulane system (Fig. 1), allowing the first extraction module and the CEGS module to fit onto our existing lab frame and benchtop.

We implement a two-day extraction procedure with the PCEGS similar to those of Lifton et al. (2015) and Goehring et al. (2019), utilizing a lithium metaborate (LiBO$_2$) flux to dissolve the quartz and release the *in situ* $^{14}$C at 1100°C. The first day's procedures involve degassing the LiBO$_2$ flux and preparing the purified quartz for extraction, while the second day is the extraction/purification/graphitization procedure. Once started, the Day 1 LiBO$_2$ degassing process operates on the selected extraction module (either TF1 or TF2) completely independently of the main CEGS module. The Day 2 process, on the other hand, requires control from the main CEGS module to allow sample collection, purification, measurement, dilution, and graphitization. In practical terms, we execute a Day 1 process on one extraction module, then the next day start a Day 1 process on the second extraction module. The Day 2 process for the first extraction module can then be run without interruption from the main CEGS module. The Day 1 and Day 2 processes are then subsequently cycled between the two extraction modules. This comfortably allows for PCEGS throughput of 4-5 samples per week.

On Day 1, a quartz sample is pretreated with 50% (v:v) HNO$_3$:18 MΩ water for at least 90 min in an ultrasonic bath, rinsed thoroughly in 18 MΩ water, then dried in a vacuum oven overnight. A sample boat (either single-use high-purity Al$_2$O$_3$ or reusable 90%Pt-10%Rh) containing ca. 20 g of pre-fused LiBO$_2$ beads (melting point 845°C) is placed inside a flame-cleaned fused quartz sleeve in the mullite furnace tube (with borosilicate glass o-ring ball joint end seals), using flame-cleaned implements. The 24-inch-long (60.96 cm) quartz sleeve (replaced after every sample) extends beyond the furnace hot zone, protecting the furnace tube from LiBO$_2$ vapors that evolve from the fused sample at high temperature. The aggressively reactive vapors etch the interior of the sleeve within the hot zone of the furnace, instead of the furnace tube itself (Fig. 3). To prevent atmospheric CO$_2$ or other contaminants from entering the furnace tube when it is opened, the tube is first backfilled with Research Purity He (99.9999%) to 20 torr above ambient atmospheric pressure. The He is then slowly bled through the tube while open to atmosphere. Once closed again, the furnace tube is evacuated to <5 x 10$^{-3}$ torr, isolated, and 50 torr of RP O$_2$ is subsequently added. The furnace is then heated to the extraction temperature (typically 1100°C for 1 hour while O$_2$ is bled through with a mass flow controller and automated metering valve to maintain the tube pressure and to flush out any evolved contaminants to the vacuum pump. The tube is then cooled and evacuated overnight.

On Day 2, approximately 5 g of the pretreated quartz sample is evenly distributed over the now-solid LiBO$_2$ in the boat and returned to the furnace, evacuated to <5 x 10$^{-3}$ torr, isolated, and 50 torr RP O$_2$ is added again. The sample is then heated to 500°C for one hour to combust and remove atmospheric/organic contaminants, while bleeding O$_2$ across the sample as before and exhausting to the vacuum system. After that hour, the 500°C tube furnace is evacuated to <5 x 10$^{-3}$ torr. Subsequently, 50 torr of RP O$_2$ is admitted into the tube furnace and the sample/flux is



heated to 1100°C and held at the high temperature for three hours to dissolve the quartz and release any trapped carbon species. During extraction the $O_2$ pressure in the tube typically rises to ca. 60 torr.

After the extraction procedure completes, the evolved gases are bled with RP $O_2$ through the secondary oxidation furnace to ensure any carbon species released during extraction are completely oxidized to $CO_2$ before collection in the compact coil trap cooled with LN. During this step, the tube pressure at the end of the extraction step is

maintained during the bleed (to prevent excess $LiBO_2$ vaporization) while the furnace cools to <800°C (to ensure complete melt solidification), before shutting off additional $O_2$ inflow and slowly evacuating all tube gases through the secondary furnace and coil trap. The condensed gases are then transferred to the purification section to remove water, halogens, and nitrogen and sulfur oxides. The gas is transferred cryogenically with LN first into the variable temperature trap (VTT) and the incondensable gases are evacuated. The VTT is then warmed to -145°C for 10

minutes, while the evolved $CO_2$ is passed through a Cu mesh/Ag wool trap held at 600 °C and frozen with LN into the measurement chamber (MC) (Fig. 1). The $CO_2$ yield is then measured manometrically as equivalent mass of C (μg), and typically diluted to ca. 300 μg C with $^{14}C$-free $CO_2$. A ca. 9 μg C split is collected in a pre-evacuated Exetainer® vial for stable C isotopic analysis offline, and the remaining sample is transferred cryogenically to one of the six graphite reactors (Fig. 1). The sample then undergoes hydrogen reduction (with Research Purity $H_2$ –

99.9999%) to filamentous C (graphite) on an Fe catalyst, with water trapped by $Mg(ClO_4)_2$ (Southon, 2007; Santos et al., 2004). Procedural background samples are run after approximately every 7-10 unknown samples, using identical procedures without adding quartz.

Finally, the graphite is packed into an Al cathode for $^{14}C$ measurement by Accelerator Mass Spectrometry (AMS) at PRIME Lab. Sample $^{14}C/^{13}C$ ratios are measured relative to Oxalic Acid II (NIST-4990C). Stable carbon isotopic

ratios were measured at the University of California at Davis Stable Isotope Facility (stableisotopefacility.ucdavis.edu) using isotope ratio mass spectrometry (Lifton et al., 2015). Measured *in situ* $^{14}C$ concentrations are calculated from the resulting $^{14}C/C_{total}$ after subtracting representative procedural background $^{14}C$ values, following Hippe and Lifton (2014). Measurement uncertainties are presented at the 1σ level unless otherwise noted.

**3 Initial Experiments**

Once the PCEGS was operational, we began to characterize its performance in terms of procedural blank (background) values as well as measurements of intercomparison materials such as CRONUS-A (Jull et al., 2015). We also characterized the mass-dependence of graphitization blanks. Since publication of Lifton et al. (2001), we and other labs using $LiBO_2$ for extraction (e.g., Goehring et al., 2019; Lamp et al., 2019; Fülöp et al., 2010) had used

single-use high purity sintered $Al_2O_3$ combustion boats for our flux + samples. On the other hand, laboratories that implemented flux-free *in situ* $^{14}C$ extractions have either used Pt (e.g., Hippe et al., 2009, 2013; Lupker et al., 2019) or fused quartz vessels (Fülöp et al., 2015, 2019). The labs using flux-free processes typically report blanks on the order of 1-3x10$^4$ $^{14}C$ atoms (e.g., Lupker et al., 2019; Fülöp et al., 2019), while the labs using flux-based extractions have reported blanks on the order of 1-2x10$^5$ $^{14}C$ atoms (e.g., Lifton et al., 2015; Goehring et al., 2019; Lamp et al.,

2019). Goehring et al. (2019) deduced that the differences in $^{14}C$ backgrounds between the flux and flux-free



extraction systems lay at least in part with the sintered $Al_2O_3$ boats reacting with the flux to release small and variable amounts of persistent contaminant [14]C during the extraction process. They described assessing boats of alternate construction, and reported a promising process blank result of ca. $4x10^4$ [14]C atoms from an initial experiment with a reusable 90%Pt:10%Rh alloy boat.

Our initial PCEGS experiments utilized the single-use $Al_2O_3$ combustion boats, but in the meantime, we also obtained a set of 90%Pt:10%Rh (hereafter Pt/Rh) combustion boats from Heraeus Precious Metals North America LLC (www.pt-labware.com). We thus compared results using both types of boats, for both blanks and intercomparison samples. The solidified $LiBO_2$+sample melt is cleaned from the Pt/Rh boats between samples by overnight ultrasonication at 40°C in 10% (v:v) reagent grade $HNO_3$:18 MΩ water in sealed 1L polypropylene

bottles, followed by thorough rinsing in 18 MΩ water and drying in a gravity oven.

### 3.1 Graphitization blanks

The mass-dependence of the PCEGS graphitization blanks was assessed by graphitizing aliquots of [14]C-free $CO_2$ in masses ranging from ca. 50 µg C to 1000 µg C (Table 1). As with previous studies using either Zn or $H_2$ as the reducing agent for $CO_2$ to C (e.g., Donahue et al., 1990; Lifton et al., 2001, 2015; Goehring et al., 2019), we observe

an inverse relationship between sample mass and the measured [14]C/$C_{total}$. This relationship is well-characterized by the equation (adjusted $R^2 = 0.994$)

$$B_g = (1.243 \pm 0.045) \times 10^{-13} \big/ mass + (1.301 \pm 0.050) \times 10^{-15} \qquad (1)$$

Correction of the measured [14]C/$C_{total}$ for the graphitization blank ($B_g$) follows Eq. 6 of Donahue et al. (1997).

### 3.2 Procedural blank comparison

Initial experiments with the new system involved procedural blanks with our original single-use $Al_2O_3$ boats in concert with measurements of intercomparison materials (Section 3.3). Subsequently, we switched to reusable 90%Pt/10%Rh sample boats, with associated measurements of procedural blanks and intercomparison materials for a range of experimental conditions.

### 3.2.1 $Al_2O_3$ boats

The first set of blanks and intercomparison samples processed on the PCEGS with $Al_2O_3$ boats involved a more aggressive than normal Day 2 combustion step to more thoroughly remove any potential organic C that might remain on the etched sample grains. This was motivated by Nichols and Goehring (2019) who found evidence of modern [14]C contamination by laurylamine used in froth flotation mineral separation techniques that was not removed completely by their original etching procedure. Although we had not observed evidence of this issue with

*in situ* [14]C results from our lab, we tested a low-temperature combustion procedure of 2 hr. at 600°C, reasoning that Hippe et al. (2013) utilized a 2 hr. at 700°C combustion step with no apparent demonstrable effects on their results relative to combustions for 1 hr. at 500°C. This more aggressive combustion step was then followed by our normal 1100°C flux fusion for 3 hr.



Initial procedural blank experiments largely utilized TF1, and progressively increased from ca. 6.50 x $10^4$ to 1.03 x $10^5$ $^{14}$C atoms with a mean of (8.79 ± 1.64) x $10^4$ $^{14}$C atoms, while a single blank from TF2 yielded ca. 1.14 x $10^5$ $^{14}$C atoms (Table 2, Fig. 4). The source of the time-dependent increase was not identified before switching to the Pt/Rh boats, but these values still represent an improvement over blank values presented in Lifton et al. (2015) by ca. 30-70%.

### 3.2.2 Pt/Rh boats

On switching to the Pt/Rh boats, we also reverted to our original procedure utilizing a 500°C combustion step for 1 hr. It was immediately obvious that the Pt/Rh boats heat much more uniformly than the $Al_2O_3$, based on dramatic differences in the flux's corrosive effects on the quartz sleeves between the two types of boats (Fig. 3). The sleeves used with the $Al_2O_3$ boats were corroded mainly above and below the boat, as well as at the ends of the heated zone where the $LiBO_2$ vapor condenses in ca. 5 cm-wide bands (Fig. 3a). The rest of the heated portion of the sleeve is

only lightly corroded and remains transparent. However, when using the Pt/Rh boats, the $LiBO_2$ evenly corrodes the sleeve interior over the entire hot zone length (Fig. 3b). It thus appears that the more efficient heat conduction of the metal boats leads to more aggressive heating of the flux and sample than in the $Al_2O_3$ boats. Experiments with the Pt/Rh boats at extraction temperatures of 1000°C and 1050°C resulted in significantly less corrosion of the sleeve than at 1100°C (Fig. 3b).

Initial procedural blanks using the Pt/Rh boats were dramatically lower than those using the $Al_2O_3$ boats, with much better reproducibility, averaging (4.08 ± 0.66) x $10^4$ $^{14}$C atoms (1σ) (Table 3, Fig. 4). Different combinations of combustion (500°C and 600°C – 1 hr) and extraction temperatures/times (1100°C – 3 hr, 1000°C – 3 hr and 4.5 hr) were investigated as well (Table 3) (corresponding to intercomparison experiments described in Section 3.3), with no significant effect on blank results. This supports the hypothesis of Goehring et al. (2019) that a significant

component of $Al_2O_3$ procedural blanks derived from the sintered ceramic boats themselves. The improved blank reproducibility using the Pt/Rh boats allows us to identify background signals that previously we were unable to resolve. After this initial set of analyses depleted most of the bottle of Ultra-Pure grade $LiBO_2$ (Claisse C-0611-00, Batch C-10001), we switched to a new bottle of Pure grade $LiBO_2$ (Claisse C-0610-00, Batch C-17000-10). We reasoned that Pure and Ultra-Pure grades only differ in metal impurity content – both are pre-fused, spherical beads

and thus should be essentially functionally equivalent for our application.

However, subsequent blanks with the new bottle increased in both $CO_2$ yield (ca. 1.5 µg to ca. 4 µg C-equivalent) and $^{14}$C content (ca. (1.51 ± 0.31) x $10^5$ $^{14}$C atoms) (Table 3, Fig. 4). Although these values were higher than the initial measurements, they were reproducible on both TF1 and TF2, so we continued with normal system operation. Subsequently, though, the $CO_2$ yields and $^{14}$C content inexplicably jumped again to new 'stable' values of ca. 6.6 µg

C-equivalent and (2.66 ± 0.07) x $10^5$ $^{14}$C atoms, respectively, using TF1, with a similar but slightly lower result with TF2. At that point we tested a second bottle of Pure grade $LiBO_2$ (Claisse C-0610-00, Batch C-19000-10 – purchased at the same time as Batch C-17000-10) on procedural blanks in TF1 (PCEGS-94) and TF2 (PCEGS-95), with even higher results of (3.21 ± 0.10) x $10^5$ and (3.63 ± 0.15) x $10^5$ $^{14}$C atoms, respectively. The higher blanks



from Batch C-19000-10 also exhibited higher $CO_2$ yields (ca. 8-9.5 μg C-equivalent). In fact, the $CO_2$ yields from each extraction module tracked the $^{14}C$ atoms quite linearly for all these experiments, with similar regression fits to each ($R^2$ values of 0.955 and 0.970 for TF1 and TF2, respectively – Fig. 5)

At that point we paused normal system operations and conducted more basic experiments to try to isolate the source of the increased blanks – was it in the system overall or the $LiBO_2$? Two procedural blanks with everything except for the $LiBO_2$ (boat-sleeve only) – one boat cleaned in 10% v:v HCl, and the other in 10% v:v $HNO_3$ – both yielded

ca. $2.0 \times 10^4$ $^{14}C$ atoms. This indicated that the $LiBO_2$ was the source of the high blank, although the nature of that source and why the blank increased with time is unclear.

We then obtained a new bottle of Ultra-Pure grade $LiBO_2$ (Claisse C-0611-00, Batch C-19001-10); two blanks from that bottle from TF2 (PCEGS-98 and 99) yielded values comparable to PCEGS-95 – ca. $3.6-3.7 \times 10^5$ $^{14}C$ atoms, and ca. 8-9 μg C-equivalent yields (Fig. 4, Table 3). Finally, we tried a blank with the remainder of the original bottle of

Ultra-Pure grade $LiBO_2$ (Claisse C-0611-00, Batch C-10001). This experiment (PCEGS-100) exhibited $CO_2$ yield and $^{14}C$ content comparable to our original tests: 2.2 μg C-equivalent and $4.76 \pm 1.12 \times 10^4$ $^{14}C$ atoms. In consultation with Claisse technical support, we were unable to identify any chemical change in their product or manufacturing process that could have led to the progressively increasing blanks. As such, we identified another vendor, SPEX CertiPrep. We purchased a similar prefused Ultra-Pure grade $LiBO_2$ from them (FFB-0000-03, Lot

240920D-2904) and ran a blank on each extraction module. $CO_2$ yields were comparable to those of the original Claisse Ultra-Pure batch, and $^{14}C$ contents were slightly improved over that material: ca. 2.5 μg C-equivalent and ca. $3.6 \times 10^4$ $^{14}C$ atoms (Table 3, Fig. 4). Subsequent blanks with the new SPEX $LiBO_2$ were generally comparable to or better than those initial measurements, ranging from ca. $2.4 \times 10^4$ to $5.0 \times 10^4$ $^{14}C$ atoms (mean: $(3.38 \pm 0.92) \times 10^4$ $^{14}C$ atoms), and similar to recently published blank measurements from other *in situ* $^{14}C$ laboratories using Pt sample

boats (e.g., Lupker et al., 2019; Goehring et al., 2019) (Table 3, Fig. 4). Regardless of the ultimate cause of the unexplained blank behavior with the more recent bottles of Claisse $LiBO_2$, we are proceeding with the SPEX Ultra-Pure $LiBO_2$ as our preferred flux.

Late in this process we also discovered that the temperature controller for TF2 was miscalibrated at high temperature setpoints, reading 1120°C on an independent Type-S thermocouple probe when set to 1100°C. Independent

measurement of the lower temperatures for the combustion steps in TF2 agreed with the setpoints – only the extraction temperatures exhibited the offset. We subsequently adjusted the setpoint temperatures for extractions to achieve the desired temperature on that furnace (1080°C setpoint for 1100°C actual, and 1035°C setpoint for 1050°C actual). No such problem was observed with TF1. Results from both blanks and intercomparison materials (Section 3.3 below) do not appear to indicate any significant effect from the 20°C excess temperature in the affected TF2

experiments (Figs. 4 and 6, Tables 3 and 4).

### 3.3 Extraction experiments with intercomparison materials

While we worked to isolate and understand the source(s) of the time-dependent procedural blanks on our new system, we also set out to better understand the effects of different combustion temperatures/durations on the



amount of $^{14}$C extracted from the well-studied CRONUS-A intercomparison material (Jull et al., 2015). In addition,
since the more uniform heating of the Pt/Rh boats rendered the LiBO$_2$ flux more broadly aggressive toward the
fused quartz sleeves at 1100°C, we tested whether it would be possible to lower the extraction temperature and still
achieve full $^{14}$C recovery from CRONUS-A. We also initiated measurements at PRIME Lab of the *in situ* $^{14}$C
content of the CoQtz-N intercomparison material (e.g., Binnie et al., 2019) using both types of boats.

### 3.3.1 CRONUS-A – Al$_2$O$_3$ boats

Initial experiments with the Al$_2$O$_3$ boats used CRONUS-A to test whether the more aggressive combustion
procedure described in Section 3.2.1 (2 hr at 600°C) followed by a 3 hr fusion at 1100°C might affect the measured
*in situ* $^{14}$C concentrations significantly. Results from both TF1 and TF2 yielded $^{14}$C concentrations on the order of
10% below the consensus value for the material and outside the uncertainty band (Table 4, Fig. 6), suggesting
diffusive loss of *in situ* $^{14}$C during the more aggressive low-temperature combustion step. We thus subsequently
abandoned that more aggressive procedure in favor of the original 1 hr at 500°C combustion step of Lifton et al.
(2001) (also Section 3.2.2).

### 3.3.2 CRONUS-A – Pt/Rh boats

Our efforts with the Pt/Rh boats largely focused on optimizing extraction temperature and time, again using
CRONUS-A as a benchmark (Table 4, Fig. 6). We varied combustion and extraction temperatures/durations, using
corresponding background corrections appropriate for the procedures used and allowing for the observed procedural
blank time-dependence.

The experiments with extractions for 3 hr at 1100°C and 1120°C, and 4.5 hr at 1000°C (PCEGS-44, 46, 50, 90, 104,
105, 106, 133 – Table 4, Fig. 6) yielded a mean and standard deviation of $(7.08 \pm 0.17) \times 10^5$ $^{14}$C atoms g$^{-1}$ (1σ). An
additional extraction test for 3 hr at 1000°C (PCEGS-47) yielded a $^{14}$C concentration about 8% lower than this
mean, but still within the nominal range of results in Jull et al. (2015). However, we judge this extraction as likely to
be incomplete as it is outside of the 2σ uncertainty of our mean Pt/Rh CRONUS-A analyses, and as such do not
consider this further. Another test with a combustion step of 1 hr at 600°C and a normal 1100°C extraction (PCEGS-
50) yielded a result ca. 4% below the mean above using a 500°C combustion temperature but within 2σ of that
mean, and still well within the Jull et al. (2015) range. Excluding PCEGS-50 from the mean above does not
significantly change the mean nor these conclusions. After discovering the furnace controller miscalibration for TF2,
we also tested CRONUS-A results for TF2 at 1100°C (PCEGS-105), and found them indistinguishable from the
Pt/Rh mean. Finally, given the less aggressive corrosion of the quartz sleeve from tests at 1050°C (Fig. 3b), we also
tested CRONUS-A extraction for 3 hr at 1050°C (PCEGS-106), with results indistinguishable from our overall
Pt/Rh mean (Table 4, Fig. 6). We thus have switched to a 3 hr at 1050°C extraction temperature/duration going
forward.

Our CRONUS-A results are consistent with the consensus value and range of Jull et al. (2015), $(6.93 \pm 0.44) \times 10^5$
$^{14}$C atoms g$^{-1}$, as well as with the mean of our previous results at PRIME Lab (Lifton et al., 2015): $(6.89 \pm 0.04) \times$



$10^5$ $^{14}$C atoms g$^{-1}$. In addition, these new results are consistent with recent measurements by Lupker et al. (2019), Fülöp et al. (2019), and Lamp et al. (2019) (Fig. 6). Like those other studies, they also disagree with the CRONUS-A measurements of Goehring et al. (2019), for reasons yet to be determined (Fig. 6).

### 3.3.3 CoQtz-N

Our three results for the CoQtz-N intercomparison material spanned the period discussed in this work (Table 4). An initial analysis using an Al$_2$O$_3$ boat and the more aggressive 2 hr combustion at 600°C returned a lower concentration $(2.48 \pm 0.06)$ x $10^5$ $^{14}$C atoms g$^{-1}$ than the two Pt/Rh experiments at 500°C/1100°C (TF1) and 500°C/1120°C (TF2), which agree within 1σ measurement uncertainties and yield a mean value of $(2.62 \pm 0.04)$ x $10^5$ $^{14}$C atoms g$^{-1}$. Interestingly, the Al$_2$O$_3$ result with the more aggressive combustion step is only about 5% lower than the Pt/Rh mean CoQtz-N result (uncertainties overlap at 2σ), vs. 9% lower than the nominal value for the Al$_2$O$_3$ analyses of CRONUS-A. The source of this difference is not clear, but likely reflects intrinsic differences in diffusive properties of the quartz from each sample.

We only found one other study in which *in situ* $^{14}$C had been measured in CoQtz-N (Schiffer et al., 2020), but that study provides incomplete experimental details and only a plot of concentrations vs. quartz mass without any tabulated data. The four measured values presented for 1 g of CoQtz-N appear to span concentrations ca. 3 x $10^5$ to over 4 x $10^5$ $^{14}$C atoms g$^{-1}$ – well above our measured values. The source of this discrepancy merits further investigation but currently is difficult to evaluate without complete experimental details.

### 4 Conclusions

This study details key characteristics of and procedures in use for the new *in situ* $^{14}$C extraction system at PRIME Lab (PCEGS), and presents results of initial testing of procedural blanks and intercomparison materials. We compare results using the original single-use Al$_2$O$_3$ sample boats employed since Lifton et al. (2001) with those from a new set of reusable 90%Pt/10%Rh alloy sample boats.

It is clear from these experiments that the reusable Pt/Rh boats provide distinct advantages over the Al$_2$O$_3$ boats, supporting suggestions of Goehring et al. (2019). First, the Pt/Rh boats heat much more aggressively than the sintered Al$_2$O$_3$ ceramic boats, likely leading to more uniform heating of the contents. The Pt/Rh boats also appear to reduce or eliminate a significant component of the blank variability associated with the sintered ceramics, perhaps associated with small amounts of atmospheric carbon potentially incorporated into the ceramics during manufacture. Taken together, the aggressive uniform heating and purity of the Pt/Rh alloy allow for improved analytical reproducibility, allowing robust identification of systematic influences on background signals that we were previously unable to resolve with the Al$_2$O$_3$ boats.

Using the Pt/Rh boats, we demonstrated that time-dependent increases in procedural blanks were tied directly to specific batches of LiBO$_2$ fluxes manufactured by Claisse. The time-dependence did not appear to reflect flux purity, but rather some presently unknown characteristic of the Claisse fluxes appears to have changed since the original batch we used for our early experiments. Subsequent analyses with LiBO$_2$ from an alternate supplier, SPEX



CertiPrep, yielded consistently low procedural blanks on the order of $(3.4 \pm 0.9) \times 10^4$ $^{14}$C atoms, and we have switched to that flux going forward.

We also analyzed two intercomparison materials as part of our initial experiments, to confirm compatibility with earlier results from this lab as well as from others. Using both $Al_2O_3$ and Pt/Rh boats, we focused mainly on CRONUS-A, but also made initial measurements for our laboratory of the newer CoQtz-N intercomparison material. We first tested CRONUS-A in $Al_2O_3$ boats using a more aggressive combustion procedure than typically used (2 hr at 600°C vs. 1 hr at 500°C) and found significantly lower $^{14}$C concentrations from the high-temperature extraction relative to the nominal value of Jull et al. (2015), likely due to diffusive loss during the more aggressive low-temperature step. Abandoning that aggressive procedure in favor of the shorter 500°C combustion, and switching to the Pt/Rh boats, we then explored various time-temperature combinations for the high-temperature extraction step with CRONUS-A. Results for 3 hr extractions at temperatures ranging from 1050°C to 1120°C and 4.5 hr at 1000°C yielded similar results, in agreement with the consensus value as well as with published results from most laboratories, including those using our previous extraction system (Lifton et al., 2015). On the other hand, an extraction for 3 hr at 1000°C yielded a significantly lower concentration than the other analyses in this study, suggesting incomplete extraction for those conditions. Based on these results, our preferred technique is now combustion for 1 hr at 500°C followed by a 3 hr extraction at 1050°C.

The initial analysis of CoQtz-N at PRIME Lab used the more aggressive combustion step, but displayed less diffusive loss (relative to our analyses with Pt/Rh boats) than did CRONUS-A with that procedure, suggesting variable low-temperature diffusion behavior among samples. Subsequently, internally consistent results were achieved with CoQtz-N using Pt/Rh boats, with approximately 60% lower $^{14}$C concentrations than CRONUS-A. However, additional analyses of this material from this and other labs are clearly needed to work toward a consensus value.

**Declaration of competing interests**

The authors declare that they have no conflict of interest.

**Acknowledgments**

The authors gratefully acknowledge advice and technical assistance from Marc Caffee and Tom Woodruff of PRIME Lab. Funding for this research was provided by US National Science Foundation grant EAR-1560658 and a Purdue University Laboratory and University Core Facility Research Equipment Program grant (2017-2018).

**Author contributions**

This study was conceived by NL and JW. Sample preparation and analysis was done by NL and AK. NL analyzed the data and wrote the manuscript, with contributions from AK and JW.



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



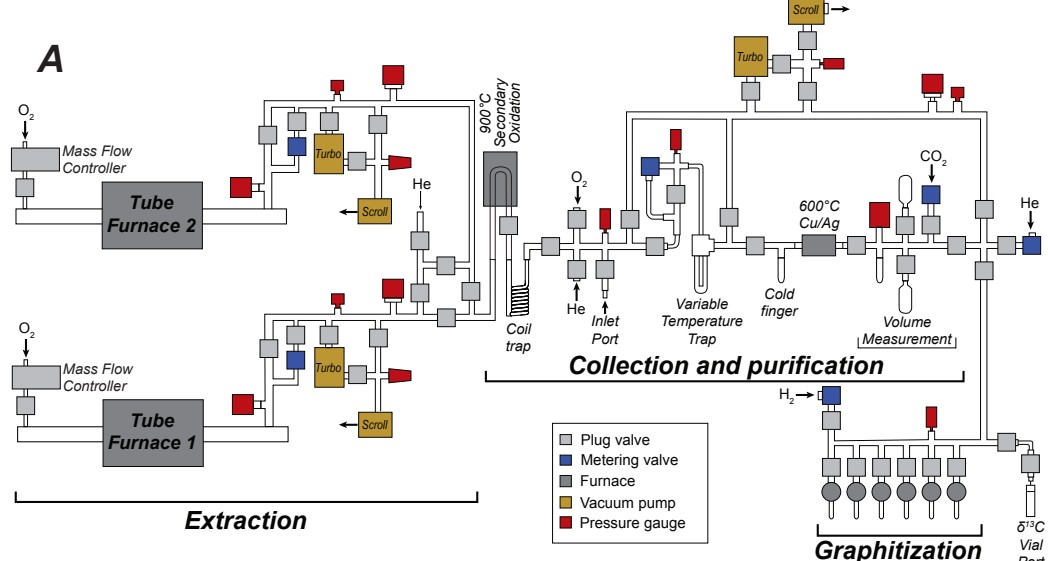

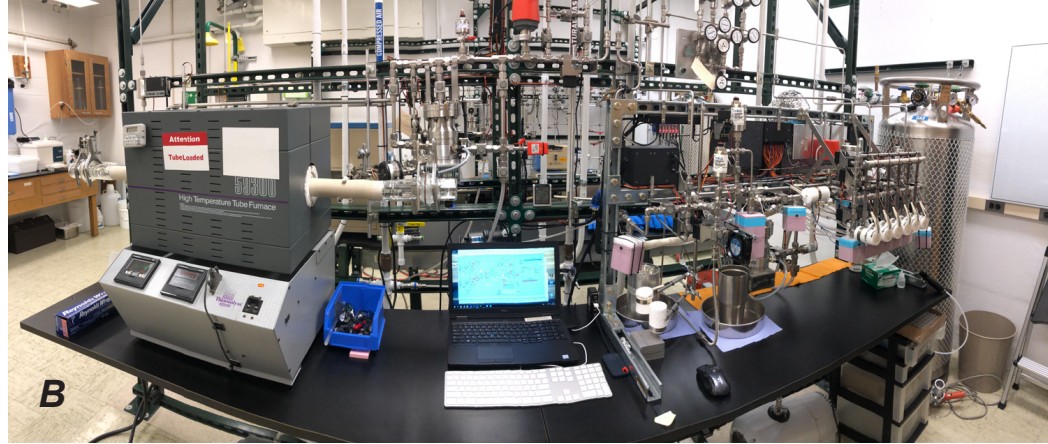


**Figure 1: A) Schematic and B) photo of the Purdue Carbon Extraction and Graphitization System (PCEGS).**



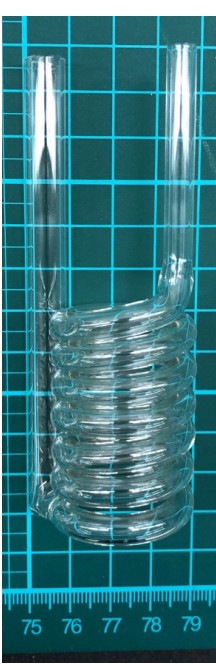

**Figure 2: Compact borosilicate glass coil trap, consisting of a 9 mm o.d. x 7 mm i.d. inlet downtube (on the left), connected to a tightly coiled 6 mm o.d. x 4 mm i.d. section with subsequent outlet tube. Scale on bottom is in cm.**



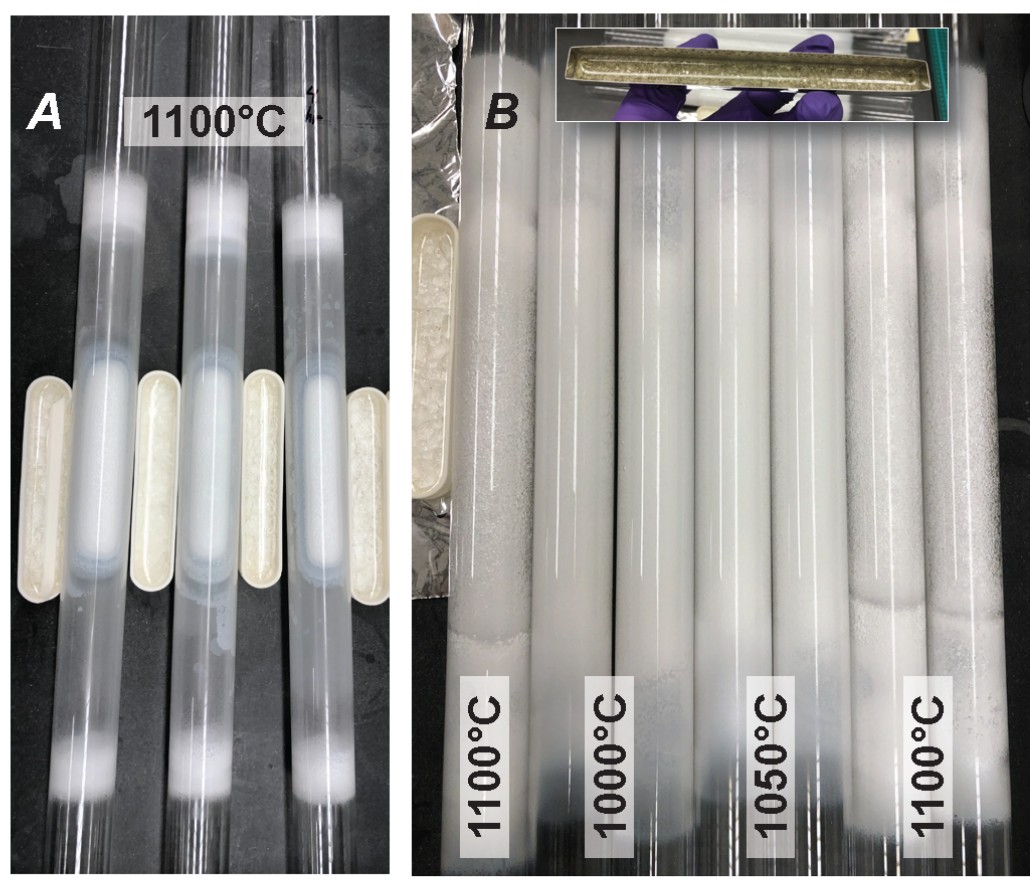

Figure 3: Comparison of quartz sleeve corrosion from $LiBO_2$ for (A) $Al_2O_3$ boats and B) Pt/Rh boats (inset), after the high temperature fusion step (3 hr) at the temperatures indicated. Note the significantly greater corrosion associated with the Pt/Rh boats vs. the $Al_2O_3$, indicating more even heating in the former, and noticeably milder corrosion from the 1050°C and 1000°C runs. The $Al_2O_3$ boat on the left side of B) is holding the sleeves in place but also serves as a comparison to A).



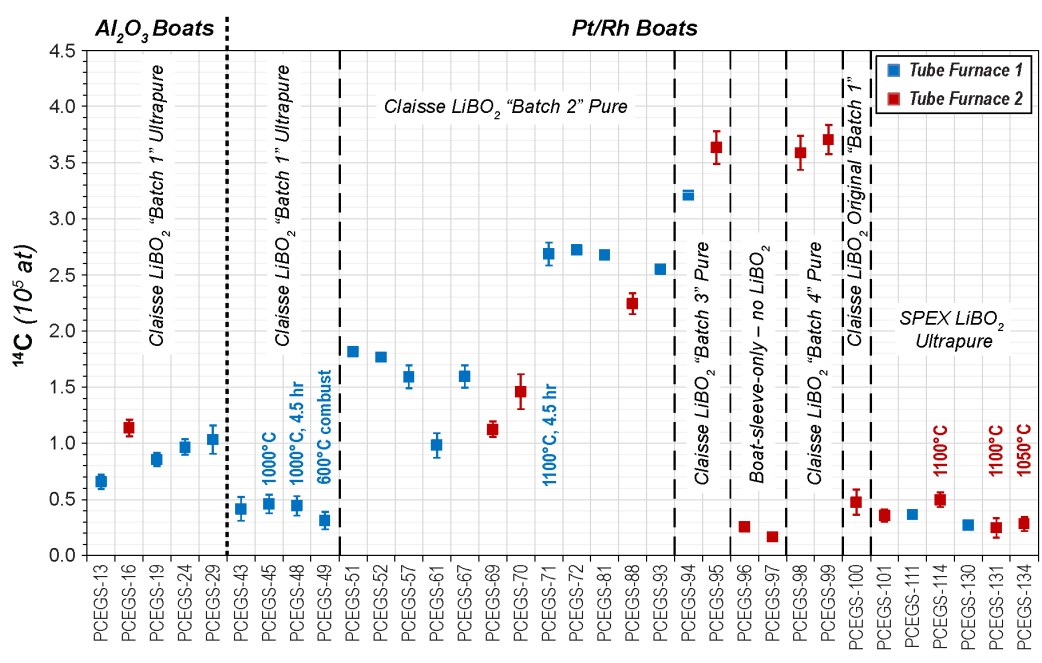


**Figure 4: Procedural blank results for Al₂O₃ and Pt/Rh boats (1σ uncertainties). All blanks using Al₂O₃ boats used a 2 hr at 600°C combustion step followed by a 3 hr extraction at 1100°C (1120°C for Tube Furnace 2 due to a miscalibration at the 1100°C setpoint). All Pt/Rh Tube Furnace 1 runs were a 1 hr at 500°C combustion step followed by a 3 hr extraction step at 1100°C, except as indicated. Tube Furnace 2 combustions with Pt/Rh boats were also 1 hr at 500°C, but**
**extractions were at 1120°C due to the miscalibration, except as indicated.**



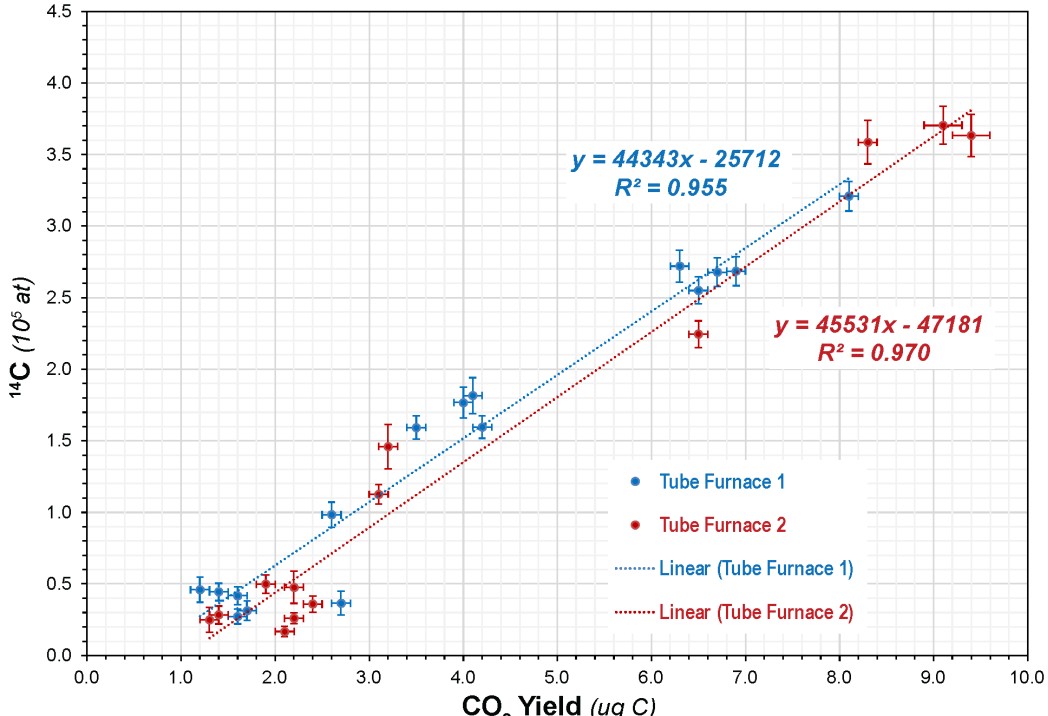

**Figure 5: Linear fit between CO$_2$ yield in µg C vs. procedural blank in $^{14}$C atoms, for Tube Furnaces 1 and 2 (1σ uncertainties shown).**




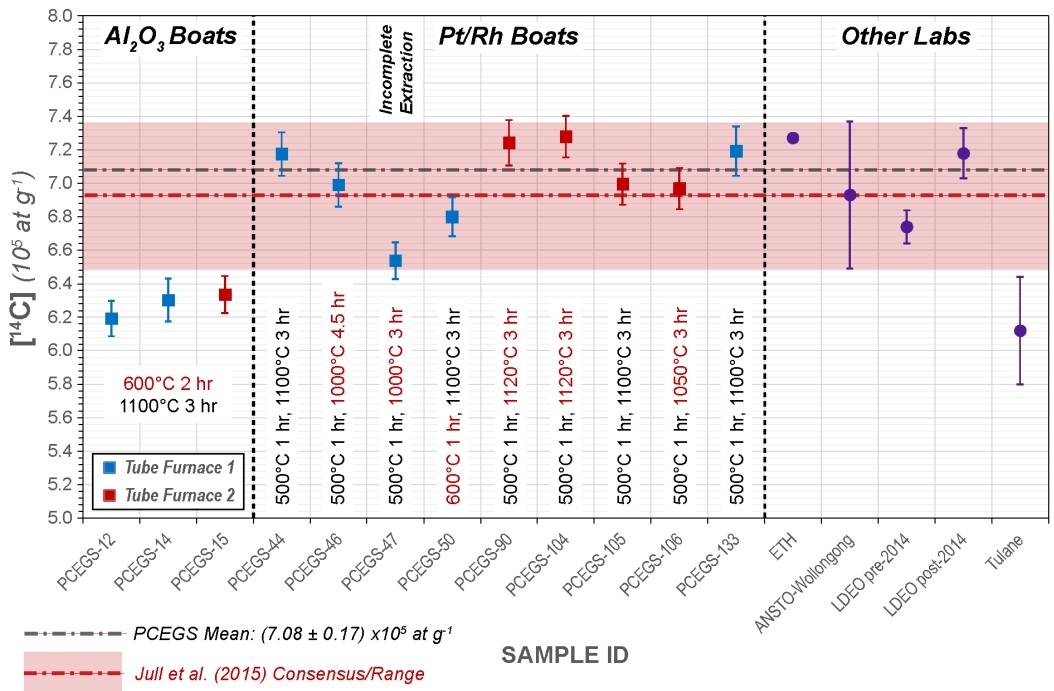

**Figure 6: CRONUS-A results with experimental details (1σ uncertainties) from this study, with mean values from other studies for comparison.**


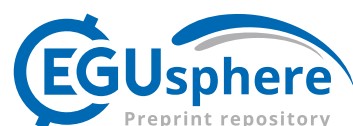

**Table 1: Graphitization Blanks**

| SAMPLE | PCEGS # | PLID[a] | Mass C $\mu g$ | $^{14}C/^{13}C$ $10^{-13}$ | $^{14}C/C_{total}$[b] $10^{-15}$ |
|---|---|---|---|---|---|
| DILGAS-300 | PCEGS-20 | 202001597 | 309.2 ± 3.8 | 1.5882 ± 0.1829 | 1.6709 ± 0.0019 |
| DILGAS-300 | PCEGS-21 | 202001598 | 339.6 ± 4.1 | 1.4773 ± 0.1817 | 1.5543 ± 0.0019 |
| DILGAS-50 | PCEGS-33 | 202100561 | 48.4 ± 0.6 | 3.7096 ± 0.4990 | 3.9024 ± 0.0053 |
| DILGAS-100 | PCEGS-34 | 202100562 | 92.6 ± 1.2 | 2.4291 ± 0.4102 | 2.5554 ± 0.0043 |
| DILGAS-200 | PCEGS-35 | 202100563 | 198.9 ± 2.4 | 1.9322 ± 0.2480 | 2.0326 ± 0.0026 |
| DILGAS-500 | PCEGS-36 | 202100564 | 523.5 ± 6.3 | 1.4752 ± 0.1897 | 1.5519 ± 0.0020 |
| DILGAS-700 | PCEGS-37 | 202100565 | 696.0 ± 8.4 | 1.4390 ± 0.2788 | 1.5138 ± 0.0029 |
| DILGAS-1000 | PCEGS-38 | 202100566 | 1,000.2 ± 12.1 | 1.4068 ± 0.1852 | 1.4799 ± 0.0020 |
| DG-05072021 | -- | 202101467 | 304.7 ± 3.7 | 2.1203 ± 0.2686 | 2.2305 ± 0.0028 |

Notes
a PRIME Lab ID
b $\delta^{13}C$ averages -45.6 ± 0.2 $\%_{oVPDB}$

**Table 2: Al$_2$O$_3$ Procedural Blanks – all used 2 hr combustion at 600°C and 3 hr extraction at 1100°C, unless otherwise noted**

| SAMPLE | PCEGS # | PLID | C yield $\mu g$ | Diluted C Mass $\mu g$ | AMS C Mass $\mu g$ | $\delta^{13}C$ $\%_{oVPDB}$ | $^{14}C/^{13}C$ $10^{-12}$ | $^{14}C/C_{total}$ $10^{-14}$ | $^{14}C$ $10^4$ atoms |
|---|---|---|---|---|---|---|---|---|---|
| | | | | | *TF1* | | | | |
| PB1-10012020 | PCEGS-13 | 202001590 | 2.8 ± 0.1 | 308.2 ± 3.7 | 299.2 ± 3.6 | -45.0 ± 0.2 | 0.5673 ± 0.0466 | 0.4256 ± 0.0494 | 6.5763 ± 0.7667 |
| PB1-10272020 | PCEGS-19 | 202001596 | 3.0 ± 0.1 | 310.4 ± 3.8 | 301.3 ± 3.7 | -45.4 ± 0.2 | 0.6857 ± 0.0357 | 0.5502 ± 0.0379 | 8.5621 ± 0.5995 |
| PB1-12032020 | PCEGS-24 | 202100567 | 4.2 ± 0.1 | 308.6 ± 3.8 | 299.6 ± 3.6 | -46.2 ± 0.2 | 0.7579 ± 0.0506 | 0.6252 ± 0.0534 | 9.6735 ± 0.8351 |
| PB1-12152020 | PCEGS-29 | 202100568 | 3.7 ± 0.1 | 303.8 ± 3.7 | 294.9 ± 3.6 | -44.9 ± 0.2 | 0.8080 ± 0.0541 | 0.6784 ± 0.0572 | 10.3334 ± 0.8796 |
| | | | | | *TF2* | | | | |
| PB2-10162020[a] | PCEGS-16 | 202001593 | 4.0 ± 0.1 | 303.7 ± 3.7 | 294.8 ± 3.6 | -45.2 ± 0.2 | 0.8731 ± 0.0460 | 0.7467 ± 0.0487 | 11.3708 ± 0.7547 |

Notes
a 1120°C extraction due to furnace miscalibration at 1100°C setpoint








**Table 3: Pt/Rh Procedural Blanks – All used 1 hr combustion at 500°C and 3 hr extraction at 1100°C, unless otherwise noted**

| SAMPLE | PCEGS # | PLID | C Mass $\mu g$ | Diluted C Mass $\mu g$ | AMS C Mass $\mu g$ | $\delta^{13}C$ ‰VPDB | $^{14}C/^{13}C$ $10^{-12}$ | $^{14}C/C_{total}$ $10^{-14}$ | $^{14}C$ $10^{\#}$ atoms |
|---|---|---|---|---|---|---|---|---|---|
| | | | | | | **TF1** | | | |
| PB1-02042021 | PCEGS-43 | 202100569 | 1.6 ± 0.1 | 300.8 ± 3.7 | 292.0 ± 3.6 | -46.3 ± 0.2 | 0.4275 ± 0.0389 | 0.2768 ± 0.0412 | 4.1740 ± 0.6240 |
| PB1-02092021[a] | PCEGS-45 | 202100571 | 1.2 ± 0.1 | 300.6 ± 3.7 | 296.0 ± 3.6 | -42.9 ± 0.2 | 0.4523 ± 0.0555 | 0.3051 ± 0.0588 | 4.5979 ± 0.8879 |
| PB1-02202021[b] | PCEGS-48 | 202100574 | 1.4 ± 0.1 | 304.1 ± 3.7 | 295.2 ± 3.6 | -46.1 ± 0.2 | 0.4407 ± 0.0374 | 0.2912 ± 0.0397 | 4.4399 ± 0.6071 |
| PB1-02232021[c] | PCEGS-49 | 202100575 | 1.7 ± 0.1 | 304.1 ± 3.7 | 295.1 ± 3.6 | -45.2 ± 0.2 | 0.3582 ± 0.0426 | 0.2047 ± 0.0452 | 3.1213 ± 0.6897 |
| PB1-03232021 | PCEGS-51 | 202101468 | 4.1 ± 0.1 | 315.8 ± 3.8 | 306.5 ± 3.7 | -43.8 ± 0.2 | 1.2488 ± 0.0737 | 1.1467 ± 0.0779 | 18.1577 ± 1.2520 |
| PB1-03252021 | PCEGS-52 | 202101469 | 4.0 ± 0.1 | 307.8 ± 3.7 | 298.8 ± 3.6 | -44.6 ± 0.2 | 1.2531 ± 0.0644 | 1.1446 ± 0.0680 | 17.6652 ± 1.0711 |
| PB1-04062021 | PCEGS-57 | 202101474 | 3.5 ± 0.1 | 304.2 ± 3.7 | 295.3 ± 3.6 | -43.9 ± 0.2 | 1.1538 ± 0.0484 | 1.0438 ± 0.0513 | 15.9198 ± 0.8064 |
| PB1-04152021 | PCEGS-61 | 202101478 | 2.6 ± 0.1 | 305.1 ± 3.7 | 296.2 ± 3.6 | -44.4 ± 0.2 | 0.7731 ± 0.0535 | 0.6423 ± 0.0566 | 9.8250 ± 0.8742 |
| PB1-04292021 | PCEGS-67 | 202101479 | 4.2 ± 0.1 | 307.8 ± 3.7 | 298.8 ± 3.6 | -44.4 ± 0.2 | 1.1447 ± 0.0466 | 1.0340 ± 0.0493 | 15.9575 ± 0.7850 |
| PB1-05252021[d] | PCEGS-71 | 202101639 | 6.9 ± 0.1 | 316.7 ± 3.9 | 307.4 ± 3.7 | -44.8 ± 0.2 | 1.7682 ± 0.0572 | 1.6911 ± 0.0605 | 26.8532 ± 1.0157 |
| PB1-06012021 | PCEGS-72 | 202101640 | 6.3 ± 0.1 | 311.8 ± 3.8 | 302.7 ± 3.7 | -45.7 ± 0.2 | 1.8171 ± 0.0637 | 1.7401 ± 0.0672 | 27.2047 ± 1.1013 |
| PB1-06192021 | PCEGS-81 | 202101649 | 6.7 ± 0.1 | 308.5 ± 3.8 | 299.4 ± 3.6 | -42.3 ± 0.2 | 1.8025 ± 0.0573 | 1.7310 ± 0.0607 | 26.7745 ± 0.9950 |
| PB1-07242021 | PCEGS-93 | 202101661 | 6.5 ± 0.1 | 306.9 ± 3.7 | 297.9 ± 3.6 | -45.2 ± 0.2 | 1.7384 ± 0.0544 | 1.6577 ± 0.0575 | 25.5091 ± 0.9372 |
| PB1-08062021 | PCEGS-94 | 202101662 | 8.1 ± 0.1 | 310.3 ± 3.8 | 301.2 ± 3.7 | -44.7 ± 0.2 | 2.1215 ± 0.0576 | 2.0626 ± 0.0608 | 32.0902 ± 1.0248 |
| PB1-10222021 | PCEGS-111 | 202102037 | 2.7 ± 0.1 | 305.0 ± 3.7 | 296.0 ± 3.6 | -45.1 ± 0.2 | 0.3905 ± 0.0514 | 0.2389 ± 0.0544 | 3.6539 ± 0.8329 |
| PB1-11192021 | PCEGS-130 | 202102056 | 1.6 ± 0.1 | 302.1 ± 3.7 | 293.2 ± 3.6 | -45.8 ± 0.2 | 0.3348 ± 0.0320 | 0.1796 ± 0.0341 | 2.7205 ± 0.5175 |
| | | | | | | **TF2** | | | |
| PB2-05112021[e] | PCEGS-69 | 202101637 | 3.1 ± 0.1 | 309.1 ± 3.8 | 300.1 ± 3.7 | -44.6 ± 0.2 | 0.8523 ± 0.0405 | 0.7260 ± 0.0429 | 11.2509 ± 0.6798 |
| PB2-05132021[e] | PCEGS-70 | 202101638 | 3.2 ± 0.1 | 307.2 ± 3.7 | 298.2 ± 3.6 | -42.6 ± 0.2 | 1.0608 ± 0.0951 | 0.9477 ± 0.1005 | 14.5966 ± 1.5576 |
| PB2-07142021[e] | PCEGS-88 | 202101656 | 6.5 ± 0.1 | 304.3 ± 3.7 | 295.4 ± 3.6 | -45.5 ± 0.2 | 1.5619 ± 0.0557 | 1.4711 ± 0.0588 | 22.4454 ± 0.9380 |
| PB2-08112021[e] | PCEGS-95 | 202101663 | 9.4 ± 0.2 | 304.9 ± 3.7 | 295.9 ± 3.6 | -45.3 ± 0.2 | 2.4222 ± 0.0865 | 2.3766 ± 0.0912 | 36.3332 ± 1.4618 |
| Boat-HCl[e] | PCEGS-96 | 202101663 | 2.2 ± 0.1 | 305.2 ± 3.7 | 296.2 ± 3.6 | -45.8 ± 0.2 | 0.3249 ± 0.0228 | 0.1697 ± 0.0246 | 2.5962 ± 0.3770 |
| Boat- HNO3[e] | PCEGS-97 | 202101669 | 2.1 ± 0.1 | 304.9 ± 3.7 | 296.0 ± 3.6 | -45.1 ± 0.2 | 0.2671 ± 0.0226 | 0.1091 ± 0.0244 | 1.6671 ± 0.3729 |
| PB2-08312021[e] | PCEGS-98 | 202101670 | 8.3 ± 0.1 | 307.3 ± 3.7 | 298.3 ± 3.6 | -45.9 ± 0.2 | 2.3763 ± 0.0903 | 2.3272 ± 0.0951 | 35.8579 ± 1.5277 |
| PB2-09022021[e] | PCEGS-99 | 202102024 | 9.1 ± 0.2 | 303.4 ± 3.7 | 294.5 ± 3.6 | -45.0 ± 0.2 | 2.4769 ± 0.0767 | 2.4350 ± 0.0809 | 37.0417 ± 1.3107 |
| PB2-09082021[e] | PCEGS-100 | 202102025 | 2.2 ± 0.1 | 306.0 ± 3.7 | 297.0 ± 3.6 | -43.9 ± 0.2 | 0.4575 ± 0.0692 | 0.3102 ± 0.0732 | 4.7589 ± 1.1240 |
| PB2-09282021[e] | PCEGS-101 | 202102026 | 2.4 ± 0.1 | 313.1 ± 3.8 | 303.9 ± 3.7 | -44.6 ± 0.2 | 0.3796 ± 0.0328 | 0.2287 ± 0.0350 | 3.5907 ± 0.5509 |
| PB2-10262021 | PCEGS-114 | 202102027 | 1.9 ± 0.1 | 302.2 ± 3.7 | 293.3 ± 3.6 | -45.4 ± 0.2 | 0.4763 ± 0.0398 | 0.3287 ± 0.0422 | 4.9805 ± 0.6425 |
| PB2-11232021 | PCEGS-131 | 202102057 | 1.3 ± 0.1 | 304.4 ± 3.7 | 295.5 ± 3.6 | -45.6 ± 0.2 | 0.3183 ± 0.0541 | 0.1627 ± 0.0572 | 2.4830 ± 0.8728 |
| PB2-12022021[f] | PCEGS-134 | 202102060 | 1.4 ± 0.1 | 301.9 ± 3.7 | 293.0 ± 3.6 | -45.9 ± 0.2 | 0.3420 ± 0.0394 | 0.1871 ± 0.0417 | 2.8326 ± 0.6326 |

Notes
a   1000°C extraction, 3 hr



b   1000°C extraction, 4.5 hr
c   600°C combustion, 1 hr
d   1100°C extraction, 4.5 hr
e   1120°C extraction due to furnace miscalibration at 1100°C setpoint
f   1050°C extraction





**Table 4: Intercomparison samples – All analyses used 1 hr combustion at 500°C and 3 hr extraction at 1100°C, unless otherwise noted**

| SAMPLE | PCEGS # | PLID | Mass Quartz g | C yield μg | Diluted mass C μg | AMS Split Mass C μg | δ13C ‰VPDB | 14C/13C 10^-11 | 14C/Ctotal 10^-13 | 14C 10^6 at | [14C] 10^5 atoms g^-1 | 14C Blank 10^4 atoms |
|---|---|---|---|---|---|---|---|---|---|---|---|---|
| *CRONUS-A* | | | | | | | | | | | | |
| *Al₂O₃[a]* | | | | | | | | | | | | |
| CRA-09172020 | PCEGS-12 | 202001589 | 5.0549 | 24.7 ± 0.3 | 306.4 ± 3.7 | 297.4 ± 3.6 | -41.6 ± 0.2 | 1.9916 ± 0.0210 | 2.0868 ± 0.0222 | 3.1303 ± 0.0535 | 6.1925 ± 0.1058 | 7.5692 ± 1.4042 |
| CRA-10072020 | PCEGS-14 | 202001591 | 5.0008 | 25.5 ± 0.4 | 377.6 ± 4.6 | 366.5 ± 4.4 | -42.9 ± 0.2 | 1.6313 ± 0.0242 | 1.7048 ± 0.0256 | 3.1519 ± 0.0639 | 6.3028 ± 0.1279 | 7.5692 ± 1.4042 |
| CRA-10132020 | PCEGS-15 | 202001592 | 5.0556 | 25.7 ± 0.4 | 303.2 ± 3.7 | 294.3 ± 3.6 | -42.2 ± 0.2 | 2.0829 ± 0.0239 | 2.1819 ± 0.0252 | 3.2033 ± 0.0562 | 6.3361 ± 0.1113 | 11.3708 ± 0.7547 |
| *Pt/Rh* | | | | | | | | | | | | |
| CRA-02062021 | PCEGS-44 | 202100570 | 5.0415 | 26.3 ± 0.4 | 303.0 ± 3.7 | 294.1 ± 3.6 | -43.1 ± 0.2 | 2.3000 ± 0.0297 | 2.4085 ± 0.0313 | 3.6174 ± 0.0656 | 7.1753 ± 0.1300 | 4.1740 ± 0.6240 |
| CRA-02112021[b] | PCEGS-46 | 202100572 | 5.0099 | 26.2 ± 0.4 | 302.4 ± 3.7 | 293.5 ± 3.6 | -42.1 ± 0.2 | 2.2316 ± 0.0296 | 2.3390 ± 0.0313 | 3.5020 ± 0.0649 | 6.9902 ± 0.1295 | 4.4399 ± 0.8879 |
| CRA-02182021[c] | PCEGS-47 | 202100573 | 5.0048 | 25.3 ± 0.4 | 303.9 ± 3.7 | 295.0 ± 3.6 | -43.4 ± 0.2 | 2.0817 ± 0.0236 | 2.1776 ± 0.0249 | 3.2722 ± 0.0558 | 6.5381 ± 0.1115 | 4.5979 ± 0.6071 |
| CRA-02252021[d] | PCEGS-50 | 202100576 | 5.0630 | 23.1 ± 0.3 | 302.4 ± 3.7 | 293.5 ± 3.6 | -42.0 ± 0.2 | 2.1861 ± 0.0251 | 2.2910 ± 0.0265 | 3.4425 ± 0.0589 | 6.7993 ± 0.1163 | 3.1213 ± 0.6897 |
| CRA-07172021[e] | PCEGS-90 | 202101658 | 5.0250 | 30.3 ± 0.4 | 309.6 ± 3.8 | 300.5 ± 3.7 | -42.6 ± 0.2 | 2.3751 ± 0.0296 | 2.4891 ± 0.0312 | 3.6395 ± 0.0685 | 7.2428 ± 0.1362 | 22.4454 ± 0.9380 |
| CRA-10072021[e] | PCEGS-104 | 202102030 | 5.0568 | 25.9 ± 0.4 | 303.5 ± 3.7 | 294.6 ± 3.6 | -42.9 ± 0.2 | 2.3318 ± 0.0268 | 2.4425 ± 0.0283 | 3.6810 ± 0.0628 | 7.2793 ± 0.1241 | 3.5907 ± 0.5509 |
| CRA-10092021 | PCEGS-105 | 202102031 | 4.7910 | 24.9 ± 0.3 | 304.5 ± 3.7 | 295.6 ± 3.6 | -43.0 ± 0.2 | 2.1205 ± 0.0240 | 2.2197 ± 0.0253 | 3.3516 ± 0.0591 | 6.9955 ± 0.1234 | 3.7317 ± 1.7660 |
| CRA-10122021[f] | PCEGS-106 | 202102032 | 4.7458 | 25.2 ± 0.4 | 306.0 ± 3.7 | 297.0 ± 3.6 | -43.1 ± 0.2 | 2.0775 ± 0.0261 | 2.1740 ± 0.0275 | 3.3071 ± 0.0587 | 6.9686 ± 0.1237 | 2.8326 ± 0.6326 |
| CRA-12012021 | PCEGS-133 | 202102059 | 5.0281 | 25.3 ± 0.4 | 303.2 ± 3.7 | 294.3 ± 3.6 | -43.8 ± 0.2 | 2.2933 ± 0.0369 | 2.3997 ± 0.0389 | 3.6163 ± 0.0743 | 7.1922 ± 0.1477 | 3.1872 ± 0.6600 |
| *CoQtz-N* | | | | | | | | | | | | |
| *Al₂O₃* | | | | | | | | | | | | |
| CQN-1022020[a] | PCEGS-18 | 202001595 | 5.0112 | 7.5 ± 0.1 | 307.2 ± 3.7 | 298.2 ± 3.6 | -44.6 ± 0.2 | 0.8281 ± 0.0133 | 0.8549 ± 0.0140 | 1.2412 ± 0.0303 | 2.4768 ± 0.0604 | 7.5692 ± 1.4042 |
| *Pt/Rh* | | | | | | | | | | | | |
| CQN-05012021 | PCEGS-68 | 2021101480 | 5.0525 | 7.0 ± 0.1 | 307.9 ± 3.7 | 298.8 ± 3.6 | -43.6 ± 0.2 | 0.9122 ± 0.0134 | 0.9444 ± 0.0142 | 1.3071 ± 0.0419 | 2.5870 ± 0.0830 | 15.1188 ± 3.1330 |
| CQN-10052021[e] | PCEGS-103 | 202102029 | 5.0289 | 6.2 ± 0.1 | 304.7 ± 3.7 | 295.7 ± 3.6 | -45.3 ± 0.2 | 0.8673 ± 0.0164 | 0.8954 ± 0.0173 | 1.3321 ± 0.0317 | 2.6488 ± 0.0630 | 3.5907 ± 0.5509 |

Notes
a   2 hr at 600°C combustion
b   4.5 hr at 1000°C extraction
c   3 hr at 1000°C combustion
d   1 hr at 600°C combustion
e   3 hr at 1120°C extraction
f   3 hr at 1050°C extraction

