# Peer review of "Technical note: Studying Li-metaborate fluxes and lowtemperature combustion/high-temperature extraction systematics with a new, fully automated *in situ* cosmogenic 14C processing system at PRIME Lab"

_EGUsphere, 2023_

## Author Comment (AC1)

We thank Dr. Schimmelpfennig for her constructive input and have done our best to address her comments, questions, and suggestions in red below.

RC1

This technical note presents the new automated in situ cosmogenic carbon-14 extraction system at PRIME lab. Improvements and advantages of the new installations compared to the previous system are described. Tests with different sample boat materials and various Lithium Metaborate flux batches show the impact of these materials on the procedural blank levels. Experiments with different preheating and extraction temperatures demonstrate the importance of these temperatures for the 14C recovery from the largely used inter-comparison material CRONUS-A and are therefore very useful for other in situ 14C labs.

The manuscript reads very well and the data is very well presented and described. Below I only list a few technical suggestions that could be considered.

The title is a bit bulky and could be clarified in my opinion. In particular I find a few terms confusing: First, it only becomes clear in the text that you use the term « combustion » to refer to the preheating step that removes atmospheric/organic C. Isn't heating to 1000-1100°C during the in situ extraction also combustion? Secondly, people who are familiar with in situ 14C extraction procedures would associate "high-temperature extraction" with systems that do not use the LiBO2 flux (heating at much higher temperatures instead). Maybe rather change the title to "Studying Li-metaborate fluxes and preheating and extraction temperatures with a new, fully automated *in situ* cosmogenic 14C processing system at PRIME Lab"?

We agree and have simplified the title to "Technical note: Studying Li-metaborate fluxes and extraction protocols with a new, fully automated *in situ* cosmogenic $^{14}$C processing system at PRIME Lab"

Lines 140-141: Please specify whether or not you also add He on Day 2 when you insert the sample.

Good point. We have clarified that section – added any time that we open the tube.

Lines 241-251: Concerning the high 14C and C blanks from the various LiBO2 batches: Isn't the heating on Day 1 supposed to degas all the CO2 in the LiBO2? Was a longer degassing duration tested to reduce the blanks? BTW, the heating temperature on Day 1 is not specified. Is it always the same the extraction temperature on Day 2?

The short answer to the first question is yes. We don't understand the mechanism for the observed higher blanks in the various batches, but clearly there is a systematic increase. We have clarified this in the text.

In terms of the longer degassing duration – the aggressiveness of the $LiBO_2$ in etching the quartz sleeve limits how long we can heat the flux in total at the extraction temperatures investigated. From much previous experience, if we heat the $LiBO_2$ for a longer time or to a higher temperature on the first day, while keeping the extraction temperature/time the same, we risk destroying the sleeve and potentially damaging the mullite furnace tube, so we are very cautious on that front. We prefer not to speculate as to whether or not such procedures might improve the blanks at this point, given that the source is clearly something associated with the Claisse products.

The degassing temperature is already specified in line 138.

Lines 290-296: It would have been interesting to test the 500°C preheating with the Al2O3 boats. Can you exclude that the 10% difference in 14C is due to absorption by the Al2O3?

In Lifton's experience, he's never seen any demonstrable absorption of $CO_2$ by the alumina boats. The previous iteration of the Purdue extraction systems in Lifton et al. (2015) yielded results with a 500 °C preheating step consistent with the consensus CRONUS-A value. We now show the Lifton et al. (2015) mean value as well on Figure 6 and cite that in the text.

Lines 303-315: For the CRONUS-A extractions in the Pt/Rh boat, there seems to be slight trend though of 14C concentrations increasing with extraction temperature: 4 of the 5 extractions at 1100°C are systematically higher than those at 1050° (3h) and 1000° (4.5h). It would be interesting to test further if there is systematic impact or not. Especially because the mean of CRONUS-A concentrations at ETH being the highest of all published values (ref your Fig. 6), was derived with a higher extraction temperature than elsewhere.

We agree with the reviewer that this could possibly be the case but are cautious in drawing too strong a conclusion from small numbers of results under variable conditions (particularly at the lower temperatures). We thus prefer to leave the text as-is on this point. We intend to continue measuring CRONUS-A and CoQtz-N with our newly established procedures going forward so hopefully more robust relationships and statistics can be derived with the additional data, as suggested by the reviewer.

Fig. 6: It would be helpful if the 1sigma band of the PCEGS mean was shown, and also to clarify on the figure if **PCEGS-47 and -50 included in that mean value**.

Added shading to the figure as suggested. Also added to the caption – the mean on the figure only drops PCEGS-47. Also dropping PCEGS-50 does not change the mean significantly, so we prefer the former number: $(7.08\pm0.17) \times 10^5$ at g$^{-1}$ (47 only dropped) vs $(7.12\pm0.13) \times 10^5$ at g$^{-1}$ (47 and 50 dropped).

Table 4: Shouldn't the first three samples get a footnote saying that they were combusted at 600°C?

Yes, the footnote is already listed below the table and the superscript is on the $Al_2O_3$ subheading. We have modified this to have the superscripts on the individual samples for clarity, per the suggestion.

Irene Schimmelpfennig

---

## Author Comment (AC2)

We thank Dr. Lamp for her constructive input and have done our best to address her comments, questions, and suggestions in red below.

RC2
In this technical note, the authors describe the new automated *in situ* cosmogenic carbon-14 (C-14) extraction system at Purdue University/PRIME Lab. The system is based on previous designs by N. Lifton and B. Goehring (formerly at Tulane) which use $LiBO_2$ flux to dissolve quartz sample grains, but upgraded with a glass coil to trap gases during sample extraction, and two extraction ovens isolated from the rest of the extraction/purification/graphitization processes, allowing for overlap. The paper presents the results of blank and interlaboratory standards (CRONUS-A, CoQtz-N) using different bottles of pre-fused $LiBO_2$ flux beads, various combustion and fusion temperatures/timings, and $Al_2O_3$ vs. Pt/Rh sample boats. The authors show that: (1) Pt/Rh boats heat more quickly and provide much lower blanks than $Al_2O_3$ boats, (2) a too aggressive combustion step results in diffusive loss of C-14, (3) bottles of flux from a certain manufacturer result in higher and increasing blanks, (4) standard measurements on the Purdue system are in line with accepted values from other laboratories.

For the most part the data contained in this technical note is presented well. I offer here some items that could be clarified for those less familiar with in situ C-14 extraction, and other suggestions/questions:

In general, I think a flow diagram of the procedures with approximate timings would be helpful. I found the discussion of "Day 1/Day 2 procedures" and degassing vs. combustion vs. extraction difficult to follow in the text.

Added a new Table 1 describing the procedural sequences on each day, and renumbered existing tables.

Lines 81-82. It would be nice to briefly describe the new CEGS $LN_2$ distribution system since the issues with the former design were discussed in detail.

The LN distribution system is described in Goehring et al. (2019) – added a citation indicating that.

Line 105: A comma after "downtube" would make this sentence clearer.

Done.

Lines 152-154: Specify which steps/traps are removing which gases.

Done.

Lines 197-198: Specify units of *Bg* (i.e., blank value in C-14 atoms or as $^{14}C/C_{total}$), and perhaps add a brief sentence on why this relationship exists.

Done.

Line 221: Does the etching pattern necessarily mean that the Pt/Rh boat heats more evenly, or just faster? The etching patten of the $Al_2O_3$ boats in Fig. 3 shows etching of the tube along the full length of the boat.

Most of the sample/boat heating likely occurs via conduction from the bottom edges of the boats in contact with the quartz sleeve. Metals conduct heat far more easily than ceramics. We speculate that the lower thermal conductivity of the $Al_2O_3$ boats may result in a thermal gradient from the base of the boat

to the top, resulting in a lower equilibrium temperature of the melt and boat than the furnace set point. Conversely, we suggest that the more rapid conduction of heat throughout the Pt/Rh boats should prevent a thermal gradient within the boat, yielding an equilibrium temperature close to that of the set point, with correspondingly higher rates of vaporization of the flux than in the $Al_2O_3$ boats. We have no information as to what the apparent temperature difference between the two types of boats may be. Regardless, it seems that the less intense pattern of etching associated with the ceramic boats is related to the thermal conductivity differences between the two materials, as all other experimental equipment was the same. We believe that the text makes this point reasonably clearly already, but we have tried to clarify further.

Line 257: These two blanks are listed as Claisse Pure ("Batch 4") on Fig. 4, but described as Ultra-Pure in the text.

Good catch – corrected on the Figure.

Lines 284-288: Add a short sentence or two describing the two quartz standards used (CRONUS-A, CoQtz-N). E.g., where they come from, approximate age/exposure history, etc.

Done

Fig. 3: Either use "(A)" or "A)" for consistency.

Done.

Fig. 4: Can the $LiBO_2$ Batch #s listed on the Figure (Batches 1-4) be updated to reflect the Batch #s/IDs used in the text (or can you list the Batch # used in Fig. 4 along with the actual bottle batch # in the text?) so the reader can more easily follow which blanks are from which bottle?

There's not enough room on the Figure to add the full Batch #s for the bottles, but we have included the informal batch numbers in the text to assist the reader in following which is which. We have also added the full and informal batch numbers to the notes in Tables 3 and 4 (updated numbering).

Fig. 6: Can you add shading around the PCEGS mean value representing the 1 std dev (similar to how the Jull et al., 2015 data is presented)?

Done.

Tables 2, 3: Can you note in the tables which blanks used which batch/bottle of flux beads?

Done – added superscripts to the PCEGS numbers and defined in the notes.